SciPost Physics

Submission

# Many-body localization in a quasiperiodic Fibonacci chain

N. Macé*, N. Laflorencie, F. Alet

Laboratoire de Physique Théorique, IRSAMC, Université de Toulouse, CNRS, UPS, 31062 Toulouse, France
* mace@irsamc.ups-tlse.fr

March 28, 2019

## Abstract

We study the many-body localization (MBL) properties of a chain of interacting fermions subject to a quasiperiodic potential such that the non-interacting chain is always delocalized and displays multifractality. Contrary to naive expectations, adding interactions in this systems does not enhance delocalization, and a MBL transition is observed. Due to the local properties of the quasiperiodic potential, the MBL phase presents specific features, such as additional peaks in the density distribution. We furthermore investigate the fate of multifractality in the ergodic phase for low potential values. Our analysis is based on exact numerical studies of eigenstates and dynamical properties after a quench.

# 1  Introduction

The question of many-body localization (MBL) aims at extending the non-interacting Anderson localization problem [1] towards more realistic systems where interparticle interactions cannot be neglected. Precursor works highlighted the possibility of a dynamical transition between ergodic and localized regimes [2–5]. This scenario is now well established for one dimensional quantum interacting systems on the lattice in the presence of disorder. A high energy transition between a delocalized phase satisfying the eigenstate thermalization hypothesis (ETH) [6,7], and a localized MBL regime with emerging integrals of motion [8,9] has been intensively studied. In such a flourishing field of research (see recent reviews [10–12]), numerical simulations of lattice models presenting a competition between interaction and disorder play a pivotal role in our understanding of the MBL problem [13]. Most of the initial studies concentrated on models with a random local potential drawn from a uniform (most of the time box-shaped) distribution as a generic representative of disorder. However, the precise form of potential chosen can have some impact as was realized in subsequent studies. For instance, a quasiperiodic potential leads to different finite-size effects around (and possibly to a different universality class for) the ETH-MBL transition [14]. Several studies suggested that slow dynamics observed in the ETH region are due to Griffiths-type regions [15, 16], which are allowed (e.g. random box distribution) or not (quasiperiodic potential) by the type of potential imposed. The fact that the potential couples to spin and/or charge degrees of freedom in a disordered Hubbard model leads to the existence of a full or partial MBL phase [17]. Finally, a discrete disorder distribution can also induce MBL [18,19], despite stronger finite size effects observed in the case of binary distributions [19, 25].

Quite remarkably, until now the MBL phase was found to be either induced by random interactions [26–29], or directly connected to an Anderson insulator [4,5]. In this work, we want to go further by studying a model that verifies neither of the two previous properties. Specifically, we consider the case of a potential that follows the Fibonacci quasiperiodic sequence, which displays critical delocalized eigenfunctions in the non-interacting limit, not found for other potentials studied so far. Concretely, we study the "standard model" of MBL, a spin 1/2 XXZ chain:

$$H = \frac{1}{2} \sum_i \left( S_i^+ S_{i+1}^- + S_i^- S_{i+1}^+ \right) + \Delta \sum_i S_i^z S_{i+1}^z - \sum_i h_i S_i^z, \tag{1}$$

which can be recast by a Jordan-Wigner transformation into a model of interacting fermions on a chain:

$$H = \frac{1}{2} \sum_i \left( c_i^\dagger c_{i+1} + c_{i+1}^\dagger c_i \right) + \Delta \sum_i n_i n_{i+1} - \sum_i h_i n_i + h_{\text{BC}}, \tag{2}$$

where $h_{\rm BC}$ encodes a boundary term that depends on the choice of boundary conditions[1]. The on-site potential terms $h_i$ are usually taken to be uncorrelated random variables drawn from a box distribution $h_i \in [-h, h]$, a binary distribution $h_i = \pm h$ (+ with a probability $p$), or to be correlated variables, quasiperiodically varying according to the Aubry-André rule $h_i^{\mathbf{AA}} = h\cos(2\pi\omega i + \phi)$, where $\omega$ is an irrational frequency ($\phi$ a random phase). In all these cases, the model is known to undergo a many-body localization transition as $h$ is increased [19,32–34,36,37]. Here, we focus on yet another choice for the potential, sometimes referred to as the *Fibonacci potential*. To the best of our knowledge this choice for the potential has never been studied in the MBL context (note however initial [20–23] and recent [24] studies on the low-energy properties of Eq. 2). In the non-interacting case $\Delta = 0$, both disordered and Aubry-André models can be Anderson localized. Free fermions subject to a Fibonacci sequence never exhibit Anderson localization however, but rather display criticality and multifractality (in a sense defined later) for any on-site potential strength $h$ [38–41]. Owing to the critical, multifractal, nature of its eigenstates and to the power-law behavior of its transport observables (discussed in Sec. 3), we can think of this model as following a line of metal-insulator transition points, as $h$ is varied.

Can this model host an MBL phase when interactions are added? One would naively be tempted to answer by the negative, given that interactions generally tend to favor delocalization [42,43], and that the non-interacting eigenstates are not localized to begin with. However, as we shall show below, the model under study presents an MBL phase, which hosts specific local features. The plan of the paper is as follows: in Sec. 2 we recall the Fibonacci sequence and present its basic properties, while Sec. 3 presents the well-known properties of the non-interacting model and in particular its multifractal properties. Moving on to the interacting case in Sec. 4, we study this model in light of various standard MBL probes, and conclude positively for the existence of an MBL transition. We then characterize more precisely the thermal and localized phases of the model, studying the eigenstates properties in Sec. 5, and dynamical properties in Sec. 6. Finally, we gather our findings and conclude in Sec. 7.

## 2   The Fibonacci chain

The Fibonacci chain under study is a chain of interacting spinless fermions [Eq. (2)] subject to an on-site binary potential ($h_i = \pm h$) varying according to the Fibonacci rule: $h_i = +h$ (resp. $h_i = -h$) if the $i^{\rm th}$ letter the Fibonacci sequence is an A (resp. a B).

**Fibonacci sequence**    The Fibonacci sequence is a quasiperiodic sequence of A and B letters. To construct it, start from the letter A, and apply repetitively the substitution rule

$$\sigma : \begin{cases} A \to AB \\ B \to A \end{cases} \tag{3}$$

to generate words of increasing length: $A \to AB \to ABA \to ABAAB \to \dots$. Notice that the length of the $i^{\rm th}$ word is the $i^{\rm th}$ Fibonacci number, hence the name of the model. The Fibonacci sequence is obtained by repeating the substitution procedure an infinite number

---

[1]In the remainder of the paper we always choose open boundary conditions (for reasons discussed in Sec. 2), in which case $h_{\rm BC} = -(n_1 + n_L)/2$ up to a constant term.

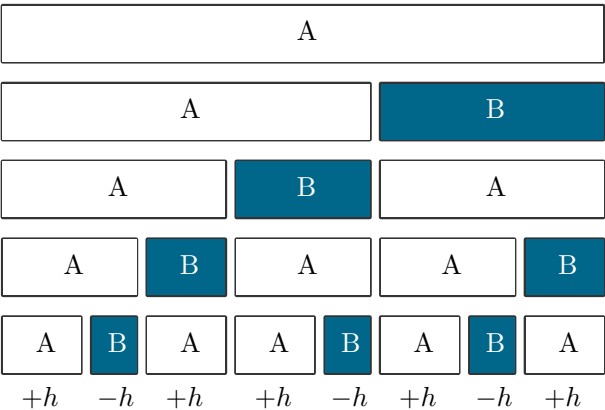

Figure 1: The first five Fibonacci words, depicted as horizontal chains of rectangular tiles. Applying the substitution rule $\sigma$ to a word yields the next, represented just below it. Below the largest word is printed the corresponding sequence of on-site potentials $\pm h$.

of times. Fig. 1 displays the first Fibonacci words. On the infinite sequence, it is easy to show that A (resp. B) letters occur with frequency $1/\tau$ (resp. $1/\tau^2$), where $\tau$ is the golden ratio, $\tau = (1+\sqrt{5})/2$. Because these frequencies are incommensurate, the Fibonacci sequence cannot be periodic. It is however close to periodicity (quasiperiodic), as we explain now.

**Quasiperiodicity, samples of finite size and averaging over realizations** When performing experiments or numerical simulations, we do not deal with the ideal, infinite Fibonacci sequence but with a chunk of length $L$. A relevant question is then: how many different words of length $L$ does one encounter in the Fibonacci sequence or, in other words: how many different samples do we have at our disposal for an experiment or a simulation? This question is crucial in practice as other potentials (e.g. the uniform box, the binary or the Aubry-André potential) all have a random component, from which a large number of different realizations can be drawn.

It turns out that there are $L+1$ distinct samples [44], each occurring with about the same frequency on the infinite chain. This makes the statistical sampling much less numerically demanding, as compared to random distributions. One can also prove [45] that a binary sequence is periodic if and only if it has strictly less than $L+1$ words of length $L$ (for $L$ larger than the period). Thus, the Fibonacci model is in some sense the closest it can be to periodicity. This also means that the Fibonacci sequence does not possess *Griffiths regions*, anomalous rare regions where the potential is markedly different from the rest of the sample. The absence of these regions is in common with the Aubry-André potential.

Among the $L+1$ samples of length $L$, one is reflection symmetric around the center of the chain. For this sample, the Hamiltonian Eq. (2) commutes with the reflection operator, and is therefore block-diagonal. As was already noted in the context of binary disorder [19], this block-diagonal structure makes such samples difficult to compare with the others. For that reason we chose to disregard them. The $L$ remaining samples (for $L$ even) all have a reflection symmetric partner. Since eigenstates and eigenspectra are identical for a sample and its partner, we are actually left with only $L/2$ truly different samples. This small number of disorder realizations naturally leads to large statistical errors[2].

---

[2]In order to alleviate the problem, we also average over energy levels within a small window for each sample.

For open boundary conditions, the reflection about the center of the sample is the only relevant symmetry operation. For periodic boundary conditions however, the conjugate action of translation and reflection can actually lead to even more equivalent samples, so that several of the $L/2$ remaining samples must be abandoned. Wishing to maximize the number of available samples, we always consider in the rest of this work systems with open boundary conditions.

# 3   The non-interacting Fibonacci chain

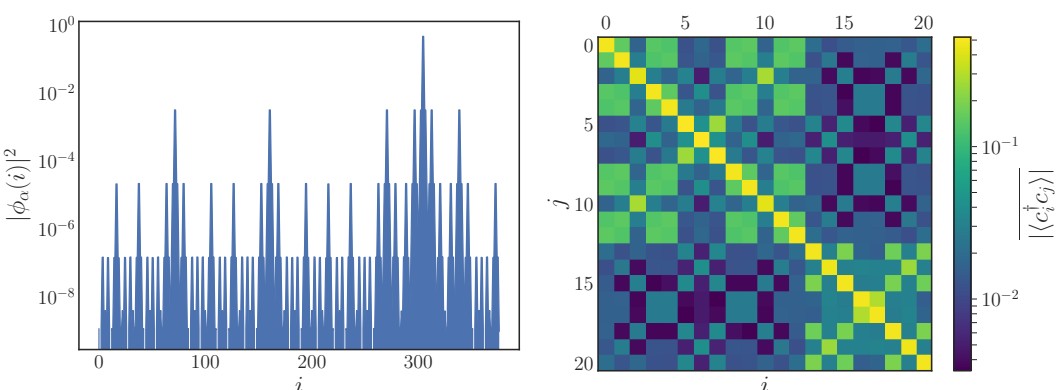

Figure 2: Two manifestations of the Fibonacci multifractality in the absence of interactions. Left: amplitude of a single-particle excitation ($h = 3$, $L = 377$), Right: average one-particle density matrix of high energy many-body eigenstates ($h = 3$, $L = 21$).

When the fermion interaction term is absent, $\Delta = 0$ in Eq. (2), the model becomes easily tractable as it can be written as $H = \sum_{i,j} c_i^\dagger \mathcal{H}_{i,j} c_j$, where the $L \times L$ matrix $\mathcal{H}$ is the single-particle Hamiltonian. Its eigenvectors, the single-particle excitations $\phi_\alpha$ and the associated single-particles energies $\epsilon_\alpha$ form the basic building blocks of the many-body physics: the many-body eigenstate of energy $\epsilon_{\alpha_1} + \epsilon_{\alpha_2} + \dots$ is the Slater determinant of the corresponding single-particle excitations $\phi_{\alpha_1}, \phi_{\alpha_2}, \dots$.

The single particle properties of the Fibonacci Hamiltonian have already been extensively studied. It was discovered that the single-particle energy spectrum and the single-particle wavefunctions are both multifractal [38–41, 46]. We recall that a wavefunction is multifractal if it is locally scale invariant. As an illustration, the left panel of Fig. 2 shows a Fibonacci wavefunction, whose peak-like structure repeats at all visible scales, hinting at its multifractality. The right panel of Fig. 2 shows the averaged one-particle density matrix of many-body eigenstates. It also exhibits a scale-invariant structure, indicating that the many-body eigenstates inherit the multifractal properties of the single-particle excitations. The density matrix of Fig. 2 (right panel) was extracted from high energy eigenstates (i.e. in the middle of the many-body spectrum), which implies that Fibonacci multifractality is very robust, surviving even at high energy. Multifractality is of course also visible in the low-energy many-body

---

This is not a solution to all problems, however, since physical quantities can be correlated from one level to the next. The gap ratios discussed in Sec. 4.2 are especially correlated since they are functions of three consecutive energy levels, and we correspondingly observe that their value is subject to particularly large fluctuations.

properties. In particular, the ground state entanglement entropy grows logarithmically with subsystem size, with quasiperiodic oscillations [47]. We conclude this brief review of the free-fermions Fibonacci chain noting that the "structural" multifractality of the spectrum and single-particle states result in a power-law behavior for the thermodynamic [22, 48, 49] and transport [50, 51] observables. For example, the conductivity at the Fermi level ($T = 0$) decays as $\sigma \sim L^{-\alpha}$ [52][3]. This behavior contrasts with the ones present for a disordered chain (exponentially decaying conductivity) and a periodic chain (no decay).

To summarize, the quasiperiodicity of the Fibonacci modulation results in the multifractality of the single-particle spectrum and wavefunctions, concomitant with power-law thermodynamic and transport observables, thus making the non-interacting Fibonacci chain an intermediate case between a periodic and a disordered system.

# 4  Many-body localization transition

Unless otherwise specified, the interaction strength is now fixed to $\Delta = 1$. We argue that the interacting Fibonacci chain exhibits a transition from a thermal to a many-body localized phase, based on the numerical study of standard observables: the level-spacing distribution [32, 53], the half-cut von Neumann entanglement entropy [54] and its variance [55]. We use in this section eigenstates in the so-called "infinite-temperature" limit, *i.e.* located in the middle of the spectrum at energy density $\epsilon = 0.5$ where $\epsilon = \frac{E - E_{\min}}{E_{\max} - E_{\min}}$ and $E_{\max/\min}$ denote the energy spectrum extrema. These eigenstates are obtained via the exact shift-invert method [13] on large chains of length $L$ up to $L = 24$. We typically consider 5000 eigenstates per sample. Dynamical probes of localization will also be discussed in Sec. 6.

## 4.1  Entanglement of eigenstates

The half-chain von Neumann entanglement entropy of an eigenstate $|\psi\rangle$ is defined as $S = -\text{Tr}(\rho_A \log \rho_A)$ where $\rho_A$ is the reduced density matrix obtained by tracing out the Hilbert space of the left/right half chain. In a thermal phase, the half-chain entanglement entropy is extensive and coincides with the thermodynamic entropy [56]: $S \sim \frac{L}{2} \log 2$ for $\epsilon = 0.5$, while in a localized phase it displays a sub-extensive area law [54]. Fig. 3 (left) shows the behavior of $\overline{S}/L$ (where the overline denotes average over samples and eigenstates) as a function of $h$, when the system size is increased from $L = 12$ to $L = 24$. For $h \lesssim 1.5$, $\overline{S}/L$ visibly tends to its maximum value while for $h \gtrsim 3.5$ it tends to zero. This is a first hint that a many-body localization transition occurs in this system. From the scaling we infer the critical potential strength to be $1.5 \leq h^* \leq 3.5$. The important fluctuations of the average entropy prevent us from obtaining a more precise bound on the location of the transition with this estimator. Let us emphasize here that this noise is not due to under-sampling, but *intrinsic* to the model: the number of available samples indeed only scales modestly as $L/2$.

The variance of the entropy divided by system size is expected to develop a peak at the MBL to thermal transition, and to go to zero away from it [55]. Following Ref. [14] we distinguish the sample-to-sample variance from the eigenstate-to-eigenstate variance within a sample. The right panel of Fig. 3 shows that both contributions indeed develop a peak around the transition. Nevertheless, the position of the variance peak strongly drifts to

---

[3]Note that the $T \to \infty$ limit is more relevant to our study. In this limit, the finite-size dependence of the conductivity has not been computed, to our knowledge.

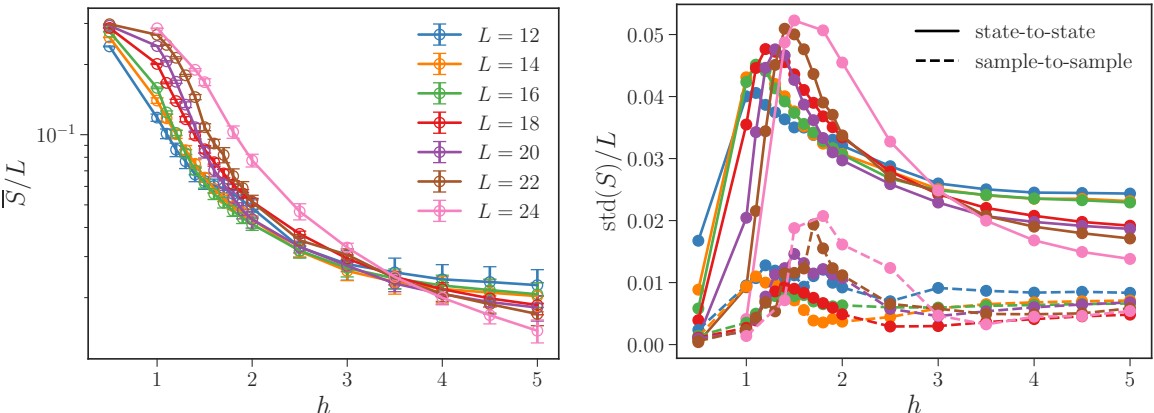

Figure 3: Left: Average half-chain entanglement entropy divided by system size, as a function of potential strength, for various system sizes. Right: Eigenstate-to-eigenstate (solid lines) and sample-to-sample (dashed lines) standard deviation of the entanglement entropy, divided by system size. Color coding for system size is the same in both panels.

larger potentials as system size is increased, a feature previously observed in other MBL systems [33,55,57]. This lets us estimate a more precise lower bound of the transition location: $h^* > 1.8$.

In uncorrelated disordered models, the sample-to-sample variance was found to be the largest [14], while here the eigenstate-to-eigenstate contribution is dominating. This inversion of the hierarchy was also previously observed in the Aubry-André model in Ref [57] where it has been attributed to the deterministic nature of the potential, preventing the appearance of Griffiths regions. The inversion observed here is consistent with this hypothesis, since the Fibonacci potential also presents this deterministic property.

## 4.2   Spectral statistics

One can also study the spectral statistics using the well-known gap ratios which measure energy level repulsion [58]. Defined as $r = \min(g_n/g_{n+1}, g_{n+1}/g_n)$, where $g_n = E_n - E_{n-1}$ is the $n^{\text{th}}$ spectral gap, they are shown in Fig. 4 for system sizes $16 \leq L \leq 24$. In the thermal phase, where energy levels follow the Wigner-Dyson statistics, they approach the value $r^{\text{Wigner-Dyson}} \simeq 0.53$ [59], while in the MBL phase they reach the lower value $r^{\text{Poisson}} \simeq 0.39$ predicted by the Poisson level statistics. In Fig. 4 we show $\bar{r}$ after averaging over 10% of the states around energy density $\epsilon = 0.5$ (except for sizes 20, 22 where 5000 states were selected, while 300 states were used for size 24). The gap ratios tend to their thermal value for $h \lesssim 1.5$, and to their MBL value for $h \gtrsim 3.5$. Large sample-to-sample variation prevents us from obtaining a more precise estimate of the transition point. Nevertheless, these results appear consistent with the behavior of the entanglement entropy. In particular they unambiguously show that an MBL phase is expected for $h > 3.5$.

## 4.3   Qualitative argument for the existence of the MBL phase

At first glance, it may seem odd that an MBL phase can exist in the Fibonacci chain, given that the free chain is never Anderson localized, contrary (to the best of our knowledge) to

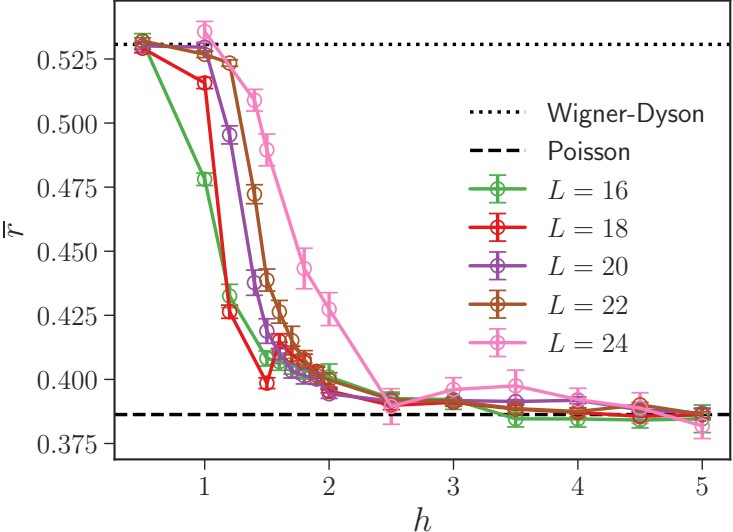

Figure 4: Gap ratio (averaged over states around energy density $\epsilon = 0.5$) as a function of on-site potential strength, for various system sizes. Gap ratio value for the Wigner-Dyson and Poisson statistics are indicated by dotted and dashed lines respectively.

all other cases of systems exhibiting MBL when adding *uniform* interactions. Note however a few examples with random interactions [26–29] as well as of interaction-induced localization in the metallic phase of two-interacting particles models in a quasiperiodic potential [30,31]. In this section, we argue that the criticality of the non-interacting Fibonacci chain is very fragile, with infinitesimal perturbations enough to drive the system into the more conventional single-particle localized or extended phases, akin to the ones of the Aubry-André model.

First, let us call $W_j$ the function that returns 0 (resp. 1) if the $j^{\text{th}}$ Fibonacci potential is $+h$ (resp. $-h$). We can Fourier transform this function [60]: $W_j = \sum_{-\infty < k < \infty} \widetilde{W}_k e^{ikj/\tau}$, with

$$\widetilde{W}_k = \frac{e^{i\frac{k}{2\tau}}}{\tau}\text{sinc}\left(\frac{k}{2\tau}\right), \tag{4}$$

where $\tau = (1 + \sqrt{5})/2$ is the golden ratio. Remark that because the Fibonacci potential is discontinuous, its harmonics decay as slow as is possible: $\widetilde{W}_k \sim 1/k$. On the opposite end of the spectrum, the Aubry-André potential enjoys infinite differentiability, and as a consequence its harmonics decays as fast as is possible (only the first is non-zero). In Ref. [61] Monthus studied the case of a free-fermion chain with a potential whose harmonics behave as $\widetilde{W}_k \sim 1/k^b$, $1 \leq b < \infty$, allowing to interpolate between the free Fibonacci ($b_{\text{Fibo}} = 1$) and Aubry-André ($b_{\text{AA}} = \infty$) chains. Ref. [61] showed that as soon as $b > 1$, there exist an extended and a localized phase. Only the free Fibonacci chain $b = 1$ is special, in the sense that it is never extended nor localized, but critical for any potential $h$.

Now, we start from the free fermions Fibonacci chain and add interactions to the picture. At the mean field level, this amounts to modifying the on-site potentials $h_i \to h_i - \Delta\langle n_{i-1} + n_{i+1}\rangle$. This modification can only smoothen the potential, hence effectively changing the exponent of the decay of the harmonics from $b_{\text{Fibo}} = 1$ to $b > 1$. According to [61], the effective free fermion model is no longer critical but instead exhibits a localization-delocalization

transition, resembling that of the Aubry-André model, which is known to exhibit an MBL phase [34–37]. We thus argue that adding interactions to the picture may have the effect of smoothening the potential, effectively pushing it away from the critical $b = 1$ line and enabling the existence of an MBL phase at large enough potential. Note also that such a scenario is consistent with recent work showing Anderson localization of free fermions with Fibonacci modulation subject to an additional weak random potential (albeit proceeding in a state-dependent non-monotonic fashion) [62].

# 5 The Fibonacci thermal and localized phases

Having established the existence of a many-body localization transition in the Fibonacci system, we now investigate in more details the content of the thermal and localized phases.

## 5.1 Local fermionic density

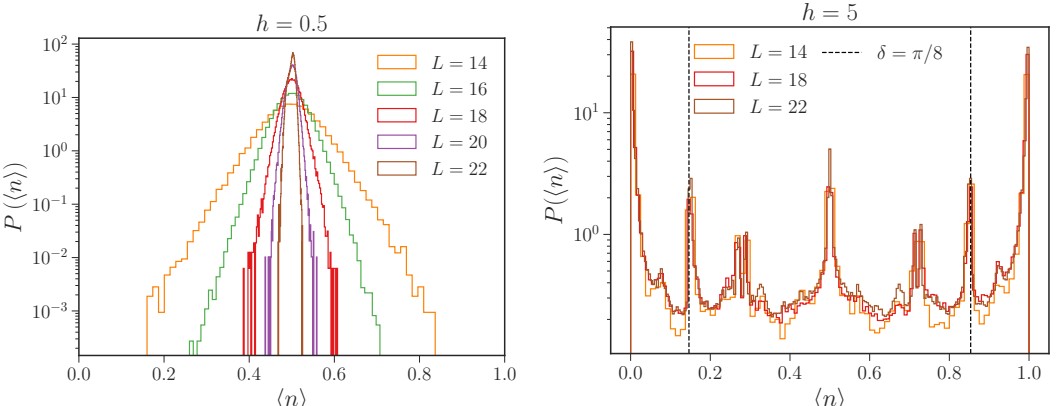

Figure 5: Left: density distribution $P(\langle n_i \rangle)$ in the thermal phase ($h = 0.5$). Right: density distribution in the MBL phase ($h = 5$). Distribution are drawn taking into account all sites and states around energy density $\epsilon = 0.5$. Dashed lines: $\langle n \rangle = \cos^2(\delta)$, $\langle n \rangle = \sin^2(\delta)$, $\delta = \pi/8$.

We start with the local density $\langle n_i \rangle$, where the average is taken over eigenstates at $\epsilon = 0.5$.

### 5.1.1 ETH phase

Fig. 5 (left) shows the local density in the ETH phase. As system size is increased, the fermionic density gets more sharply peaked at $\langle n_i \rangle = 1/2$, converging towards a Gaussian with vanishing variance for large enough system size, the expected behavior for a thermal system [63]. Physically, this simply means that the fermions are increasingly getting more homogeneously spread out over the whole chain.

### 5.1.2 MBL phase: apparition of secondary structures

The right panel of Fig. 5 shows the local density relatively deep in the MBL phase ($h = 5$) and where one can clearly see several peaks. In the large potential limit, the density operators

$n_i$ have increasingly large overlap with the local integrals of motion (commuting with the Hamiltonian) which are known to characterize the MBL phase [64]. This naturally produces symmetric peaks at $\langle n_i \rangle = 0, 1$ in the distribution of density, as observed in many simulations of MBL in spin chains (see e.g. [65]). Quite remarkably, we observe several secondary peaks which are absent in the standard MBL phase. Note also that they are not due to finite size effects.

**Numerical characterization of the secondary density peaks**    The left panel of Fig. 6 shows the site-to-site variations of the fermionic density. We observe that local density distributions are strongly correlated with the local potential landscape. In particular, on pairs of neighbouring sites subject to the same $+h$ potential, the density distribution is peaked around densities of the form $\cos^2(\delta)$ and $\sin^2(\delta)$, where the *magic angle* $\delta$ takes the values $\delta = \pi/4$ , $\pi/8$ (marked by vertical dashed lines on the right panel of Fig. 5 and on the left panel of Fig. 6). We also observe a weaker peak at $\delta = 3\pi/16$.

Deep in the MBL phase, to a good approximation, fermions are at most pairwise entangled. Then, a simple ansatz for pairs of neighboring entangled sites accounting for these magical angle densities takes the following form[4]:

$$|\psi\rangle = \cos\delta\,|01\rangle \pm \sin\delta\,|10\rangle . \tag{5}$$

We find that this ansatz reproduces better and better the features of the numerically computed one-particle density matrix $\rho_{i,j} = \langle c_i^\dagger c_j \rangle$ as $h$ and $L$ are increased. To understand why such magic angle states appear in the first place, we now turn to a toy model.

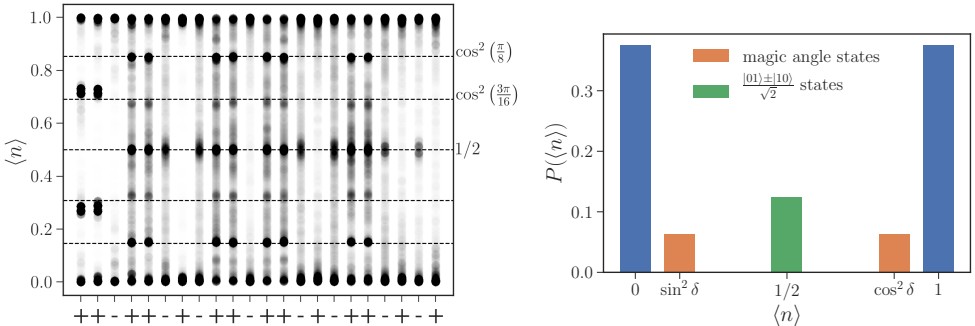

Figure 6: Left: Spatial distribution of the local fermionic density (darker means larger probability), for 5000 eigenstates at the middle of the spectrum, in the MBL phase at $h = 5$, for one of the 11 Fibonacci configurations of $L = 22$ sites. The chosen configuration is printed below the horizontal axis (+: $+h$ potential, −: $-h$ potential). Right: Density distribution for the toy model considered in the text.

**Toy model for the magic angle states.**    Let us consider a simple model of 4 fermions on a 4 sites chain. Since the magic angle densities are observed on $+h, +h$ couples (see left panel of Fig. 6), which are always surrounded by $-h$, we chose for our toy model the following sequence of on-site potentials: $\{-h, +h, +h, -h\}$. We write the XXZ Hamiltonian

---

[4]By symmetry, the state with the role of $|01\rangle$ and $|10\rangle$ exchanged is also a valid ansatz.

as $H = H_0 + H_1$ where $H_0 = -\sum_i h_i n_i$ is the dominant part deep in the MBL phase. Treating $H_1$ perturbatively, we obtain that 4 of the 16 eigenstates are magic angle states of the form (5), with

$$\delta = \arctan\left(\sqrt{1+\Delta^2} - \Delta\right). \tag{6}$$

When $\Delta = 1$ we obtain $\delta = \pi/8$ as observed numerically. Among the remaining states, 4 are $(|01\rangle \pm |10\rangle)/\sqrt{2}$ states (Eq. (5) with $\delta = \pi/4$), and the other are product states. As shown on the right panel of Fig. 6, the $\delta = \pi/4$ states give rise to the $\langle n \rangle = 1/2$ density peak, while the magic angle states give rise to the magic angle peaks. All of the three kind of states contribute to the peaks at $\langle n \rangle = 0$ or 1.

Thus, our simple 4 sites model qualitatively reproduces the observed density distribution, and predicts the correct value for the "magic angle" $\delta$. Moreover, adding more sites to the model and following the same perturbative approach, we are able to account for the other secondary peaks observed at $\delta \simeq 3\pi/16$ (visible on the left panel of Fig. 6).

Taking a step back, we observe that the existence of magic angle states stems from the binary nature of the Fibonacci potential, which causes the energy levels in the strong on-site potential limit to be highly degenerate. Hopping and interaction terms then couple states within a degenerate level, giving rise to the observed "magic angle states". In the next paragraph, we argue that these states are also favored by the correlated nature of the Fibonacci modulation.

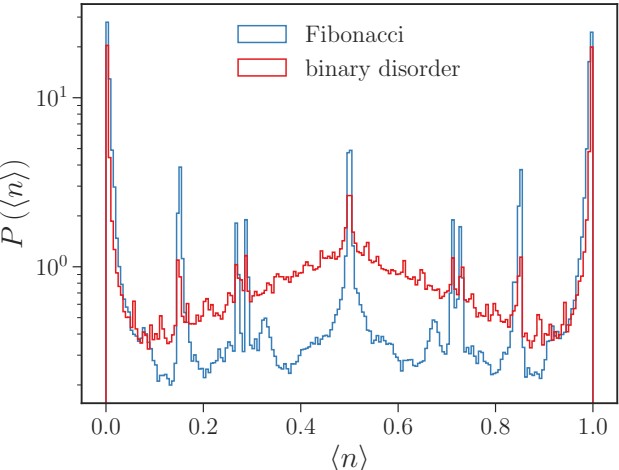

Figure 7: Comparison of Fibonacci and random binary disorder (shuffle) density distributions, for the system of $L = 22$ sites in the MBL regime at $h = 5$.

**Secondary density peaks in a random chain**   It is natural to ask whether the secondary peaks are uniquely due to the discrete (binary) nature of the Fibonacci potential or if other specificities are involved. To answer this, we should consider an on-site potential which is still binary, but no longer follows the Fibonacci sequence. A possible choice is to shuffle the Fibonacci potentials, keeping the same fraction of $\pm h$ but destroying the quasiperiodic long-range order. This is equivalent to drawing the potential from a random distribution, with $P(+h) = 1/\tau$, $P(-h) = 1 - 1/\tau$ (with $\tau$ the golden ratio). In that case we also expect a many-body localization transition [19].

The right panel of Fig. 7 compares the density distribution in the Fibonacci and random cases, for the same potential strength. The largest secondary peaks are still visible in the random case, but are lower. Moreover, the smaller secondary peaks are not visible at all. We thus conclude that the secondary peak structure is not a specificity of the Fibonacci modulation, but that the Fibonacci modulation favors the appearance of these peaks compared to the random modulation. This can be easily understood. Indeed, going back to the 4 sites toy model, one can show that the only on-site potential configurations that give rise to the magic angle states are the one allowed by the Fibonacci rule: $\{-h, +h, +h, -h\}$, and its symmetric partner obtained by exchanging $+h$ and $-h$. On the random chain considered here, these two configurations occur with a probability $2/\tau^6 \simeq 11\%$, while on the Fibonacci chain they occur at the higher frequency $(1/\tau)^3 \simeq 24\%$, due to the highly correlated nature of the Fibonacci modulation. This explains why the observed magic angle peak is less probable, but still present in the random case. To account for the other peaks observed in the Fibonacci case, we can take into account more than 4 sites (say $L$) in the toy model. While in the quasiperiodic case, there are about $L$ allowed configurations of $L$ potentials, there are exponentially more in the random case. Thus, assuming distinct configurations mainly contribute to different peaks, the probability that a given random configuration contributes to one of the Fibonacci secondary density peaks is exponentially suppressed as $L$ is increased, implying in turn that in the random case only the largest secondary peaks remain visible.

## 5.2 Entanglement entropy

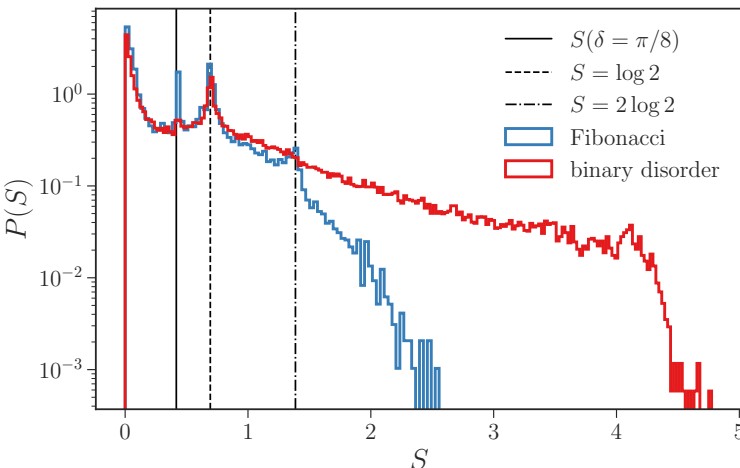

Figure 8: Distribution of the half-chain entanglement entropy, in the MBL phase ($h = 5$), with $L = 22$ sites and around energy density $\epsilon = 0.5$. For comparison, we superimpose to the Fibonacci case the distribution obtained when shuffling the Fibonacci potentials. Solid line: entropy of the "magic angle" states Eq. (5).

Fig. 8 displays the distribution of half-chain entanglement entropy, relatively deep in the MBL phase, both in the case of the Fibonacci potentials and in the case of the shuffled Fibonacci potentials (see section above). We find as expected that most eigenstates are weakly entangled, and the probability of finding large entanglement entropies decays exponentially. Besides the main peak at zero entanglement, several other peaks can again be distinguished

on the top of the background, most notably for the Fibonacci sequence. These peaks can be taken into account by again considering simple ansatz states where fermions are entangled two-by-two across the entanglement cut. Such states produce an entanglement entropy

$$S\left(\langle n\rangle\right) = -\langle n\rangle \log\langle n\rangle - (1 - \langle n\rangle)\log(1 - \langle n\rangle) \tag{7}$$

which depends on the fermion density $\langle n\rangle$ near the cut. In this approximation, the density peak at $\langle n\rangle = 1/2$ [right panel of Fig. 5], also observed in the standard MBL phase [65], contributes to the $S = \log 2$ peak of the entropy distribution (indicated by a dashed line on Fig. 8). Similarly, the "magic angle" states contribute to a peak in the entropy distribution at $S \simeq 0.42$ (indicated by a solid line on Fig. 8). In the binary disorder case, the magic angle density peak is much lower than in the Fibonacci case, and accordingly, the corresponding entropy magic angle peak is also much lower, and in fact barely visible in Fig. 8. The tail observed for $S \geq \log 2$ in Fig. 8 is due to entanglement of more than two fermions. The tail is fatter in the disordered case than in the Fibonacci case, hinting that, for the same potential strength, the Fibonacci system is more localized. Moreover, the Fibonacci entropy distribution is peaked at $S = 2\log 2$, the maximal entropy for a system of 4 fermions. Although we spotted some regularities (this peak is observed only when the on-site potential sequence is $-h, h$ on both sides of the cut), we were not able to use the same toy model to account for this peak.

## 5.3 One-particle density matrix

The one-particle density matrix is defined for a given eigenvector $|\psi\rangle$ as $\rho_{i,j} = \langle\psi| c_i^\dagger c_j |\psi\rangle$. It was used to characterize the MBL transition [66], and was subsequently studied as a probe of the MBL physics [43, 67, 68]. The study of the one-particle density matrix is especially relevant in the free Fibonacci case since it reveals signatures of its multifractality, as we are going to discuss now.

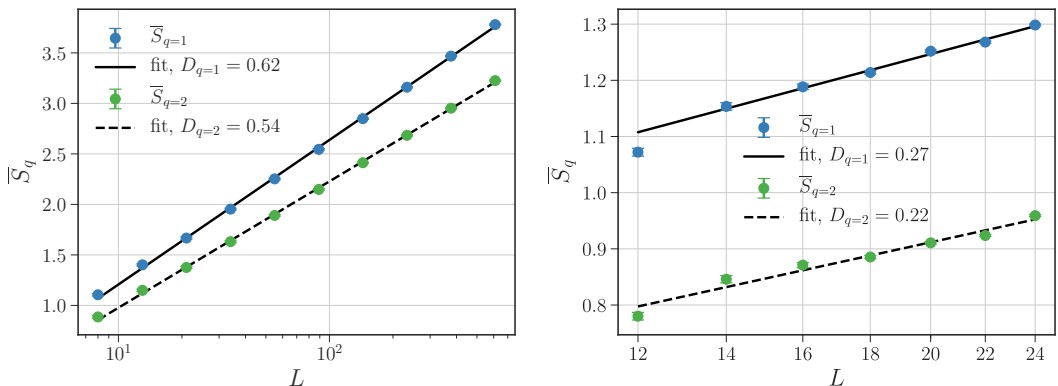

Figure 9: Scaling of the orbitals participation entropy $\overline{S}_q$ with system size, for the free (left panel) and interacting (right panel) Fibonacci chains, at $h = 1$.

**Free fermions** In the absence of interactions, as discussed in Sec. 3, the Fibonacci model is critical in the sense that single-particle excitations are multifractal, and that transport shows power-law behavior (with a continuously varying exponent). As the right panel of

Fig. 2 shows, the one-particle density matrix also bears signs of this multifractality. This is easily explained by the fact that the *natural orbitals* defined as the eigenvectors of the density matrix, coincide with the single-particle excitations. More quantitatively, calling $\phi_\alpha$ the natural orbital associated with the *occupation number* (eigenvalue of the density matrix) $n_\alpha$, we introduce the participation entropy $S_q$ of $\phi_\alpha$[5]:

$$S_q(\phi_\alpha) = \frac{1}{1-q} \log \left( \sum_{i=1}^{L} |\phi_\alpha(i)|^{2q} \right).$$
(8)

The participation entropy is a measure of how much of the realspace volume the single-particle orbitals occupy [69]. In particular, the $q = 2$ entropy is related to the inverse participation ratio (IPR): $S_2 = -\log \text{IPR}$. In general, the entropy scales as $S_q \sim D_q \times \log L + b_q$, with $0 \leq D_q(\phi_\alpha) \leq 1$ the *multifractal dimension* of the orbital $\phi_\alpha$. $D_q = 1$ signals that the orbital occupies the whole volume of the system. It is said to be *extended*. $D_q = 0$ signals a completely *localized* orbital, that is almost zero everywhere except on a finite number of sites. In the intermediate case $0 < D_q < 1$, the orbital occupies an infinite number of sites but a vanishing fraction of the volume. It is said to be *multifractal*. In the following, we study the scaling of the participation entropy averaged over orbitals:

$$S_q = \frac{\sum_{\alpha=1}^{L} n_\alpha S_q(\phi_\alpha)}{\sum_{\alpha=1}^{L} n_\alpha}.$$
(9)

The left panel of Fig. 9 shows the scaling of the participation entropy, averaged over samples and high energy many-body eigenstates, in the free Fibonacci case, with potential strength $h = 1$. As expected, we observe non-trivial fractal dimensions $0 < D_q < 1$. The multifractal dimensions vary continuously with $h$, starting from $D_q(h = 0) = 1$ and decreasing to $D_q(h \to \infty) = 0$.

**Interacting fermions** In the MBL phase, the Hamiltonian writes as a sum of commuting localized degrees of freedom (the l-bits), and thus the natural orbitals must also be localized, as discussed in [43,68]. In the ETH phase however, we rather expect the natural orbitals to be extended or multifractal, even though we are not aware of a precise generic prediction on their nature. In the case of the disordered XXZ chain, Ref. [66] argues that the natural orbitals are extended in the ETH phase, based on numerical computations. The right panel of Fig. 9 shows the scaling of the orbitals' entropy with system size. The scaling is compatible with a multifractal nature of the natural orbitals. We find multifractal dimensions substentially lower than in the free case at the same potential strength. A possible explanation is as follows: while in the free case $D_q$ vanishes only in the infinite potential strength limit $h \to \infty$, in the interacting case $D_q$ must vanish in the MBL phase, i.e. for some finite potential strength $h^*$. We indeed find (data not shown) that the participation entropy of the natural orbitals is essentially constant with system size (with small fluctuations at strong disorder strength) in the MBL phase.

In conclusion, the natural orbitals are multifractal in the free Fibonacci case, as a direct consequence of the multifractality of the single-particle excitations. In the interacting case, we find the natural orbitals to remain multifractal in the ETH phase ($h = 1$), at least up

---

[5]Notice that the entropies introduced in [43, 66] differ slightly from the ones we use here. However, we expect the scaling to be the same.

to the numerically accessible length scales. This hints that some signatures of the free chain multifractal character remain present in the ETH phase. Of course, we cannot exclude that the observed multifractal behavior is a finite-size effect and that it disappears on longer length scales (this scenario is difficult to test as we are already studing the largest possible chains). One possibility to confirm the persistence of multifractality in the interacting Fibonacci chain would be to study transport properties, which bear signatures of the Fibonacci multifractality in the non-interacting case [51].

# 6  Dynamical probes of many-body localization

In the MBL phase, part of the information contained in an initial state remains measurable at arbitrary large times, in contrast with the ETH phase where any local detail about the initial state is quickly lost. Hence, quench protocols are a natural and experimentally accessible [70] way of distinguishing between MBL and ETH phases.

In this part we study the quench dynamics through the properties of the time-evolved state $|\psi(t)\rangle = \exp(-iHt)|\psi(0)\rangle$. We use initial product states of the form $|\psi(0)\rangle = c_{i_1}^\dagger c_{i_2}^\dagger \ldots |\text{vac}\rangle$, with $L/2$ excitations $i_j$, randomly chosen with the constraint that the average energy of the state is close to the energy of a thermal state at infinite temperature[6]. The time propagation is performed using Krylov-space techniques, on chains of up to $L = 24$ sites.

## 6.1  Imbalance

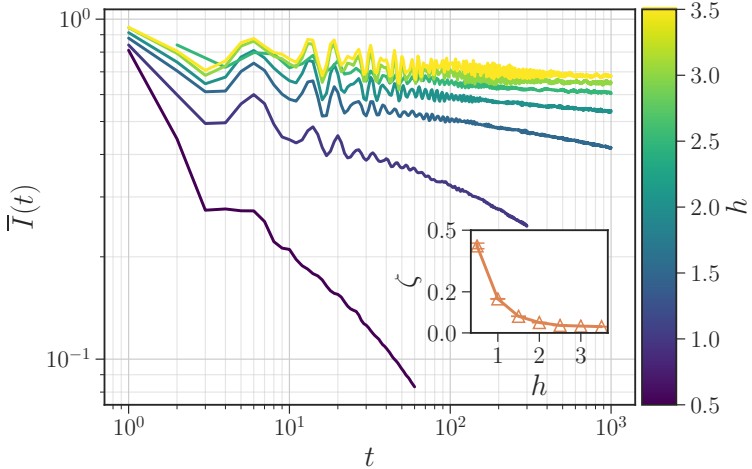

Figure 10: Imbalance as a function of time for various potential strengths, for the chains of $L = 22$ sites. Averaging is performed over initial high energy product states and field configurations. Inset: exponent of the power-law decay of the imbalance as a function of potential strength (see text).

---

[6]In practice we randomly generate product states and take the $N$ closest to the infinite temperature energy, with $N$ chosen large enough for the imbalance variance to be small.

We first examine the density imbalance (or density autocorrelation), defined as

$$I(t) = \frac{4}{L} \sum_{i=1}^{L} \left\langle \psi(0) \left| \left( n_i(t) - \frac{1}{2} \right) \left( n_i(0) - \frac{1}{2} \right) \right| \psi(0) \right\rangle. \tag{10}$$

While in the ETH phase the imbalance decays from its initial value $I(0) = 1$ to 0, in the MBL phase it saturates at a finite value, a signature that some information on the initial localization of the fermions is retained even at infinite time. Fig. 10 shows the time evolution of the imbalance, for a system of $L = 22$ sites, and for different potential strengths $h$. At low $h$, the imbalance is indeed seen to decay to 0, while at strong enough disorder ($h \gtrsim 2.5$) it saturates at a finite value. More quantitatively, at large times, we fit our data to a power-law decay of the form $I(t) \sim t^{-\zeta}$. The inset of Fig. 10 shows the exponent (extracted using the functional form given by Eq. (6) of [72]) as a function of $h$. The exponent approaches its diffusive value $\zeta = 1/2$ when $h \to 0$, and monotonously decreases as a function of $h$. It is expected to vanish (within error bars associated to the data and fitting procedure) in the vicinity of the MBL transition.

## 6.2 Entanglement growth

Starting again from a product state, the half-chain von Neumann entanglement entropy is initially 0 and increases as fermions becomes more and more entangled during the time evolution. The manner in which entanglement propagates can also be used as a probe of localization. In the ETH phase, we expect the entanglement entropy to grows quickly as $S(t) \propto t^{1/z}$ with $z = 1$ for "diffusive" [73], or $z > 1$ for subdiffusive systems [72], whereas the hallmark of the MBL phase is a slow logarithmic entanglement spreading $S(t) \propto \log t$ [74–76]. Conversely, in a (non-interacting) Anderson insulator, the entanglement entropy saturates at long time to a non thermal size-independant value.

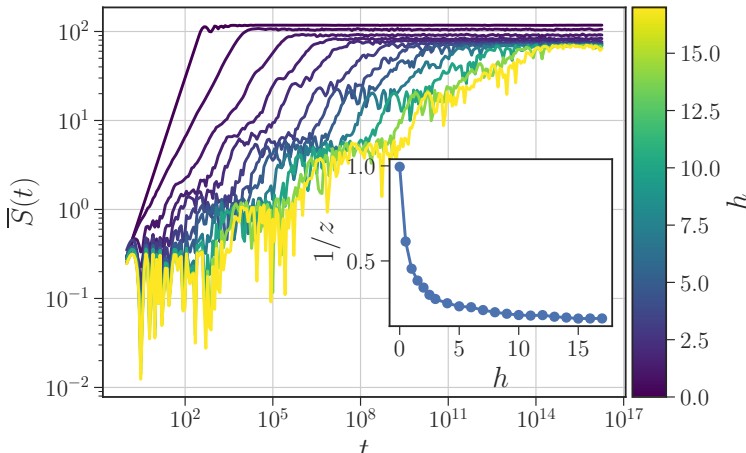

Figure 11: Free fermions entanglement as a function of time for various potential strengths, starting from random high energy initial product states on the chain of $L = 610$ sites. Inset: exponent of the power-law fit $S(t) \propto t^{1/z}$.

**Free Fibonacci chain**  In the absence of interaction, we observe (Fig. 11) that the entanglement entropy grows as a power-law: $S(t) \propto t^{1/z}$, with a monotonously increasing exponent, ranging from $z = 1$ in the clean limit ($h \ll 1$) to $z \to \infty$ in the strong quasiperiodicity limit ($h \gg 1$). This further confirms that the free Fibonacci chain is intermediate between an Anderson localized ($S(t) = \text{cst}$) and a delocalized ($S(t) \propto t$) system from the point of view of its transport properties. Remark the presence of log-periodic oscillations on top of the dominant power-law behavior. They are observed as soon as $h > 0$, and their amplitude increases monotonically with $h$. Such oscillations are the hallmark of an underlying discrete scale-invariance [77], and were observed in other dynamical quantities, such as the mean square displacement of a wave packet [78, 79]. Finally, we observe that the saturation value of the entanglement entropy slowly decreases with $h$. A scaling analysis (not shown) reveals that – as expected for a system at high energy – the entropy saturates to an extensive value, $S(t \to \infty, L) = A(h)L$, with a non-thermal prefactor $A(h)$ decreasing continuously with $h$.

**Interacting chain:  Power-law growth in the ETH phase**  At very small disorder in the ETH regime, the entanglement entropy is expected to grow linearly: $S(t) \propto t$ [73]. However, when approaching the localization transition, renormalization group theories [80,81] and effective models [15] predict that the dynamics can become subdiffusive in the presence of quenched disorder: $S(t) \propto t^{1/z}$. This subdiffusive behavior was numerically observed in the disordered case [10, 72, 82] not only close to the critical point but also deep in the ETH regime.

For the Fibonacci sequence, the upper left panel of Fig. 12 shows the entanglement as a function of time in log-log scale, for various potential strengths. At low disorder ($h \lesssim 2.5$), a clear power-law behavior is observed. The upper right panel of Fig. 12 shows the dynamical exponent $1/z$ obtained by fitting the entanglement entropy to the subdiffusive form above, for different system sizes. In an effort to free ourselves from finite-size effects, we checked (data not shown) that (i) the entanglement saturation value converges to the Page prediction for a thermalized system [56] as system size is increased, and (ii) the exponent's value does not significantly vary with system size, for large enough systems. For $h \geq 3$ we were not able to identify time scales where the entropy has a clear power-law behavior. At very low potential the exponent $z$ is close to unity, but continuously increases ($1/z$ decreases) as a function of $h$. In the vicinity of the transition point, $z(h)$ is predicted to diverge as a power-law in the case of quench disorded systems with Griffiths regions [80, 81]. Although the Fibonacci potential does not host Griffiths regions, owing to its deterministic, scale invariant nature, we observe a non-trivial $z > 1$ dynamical exponent in the vicinity of the transition point. Ref. [37, 71, 83] found a similar subdiffusive behavior using the Aubry-André Griffiths regions-free potential, even though this has been argued to be a finite-size effect [85]. Alternatively, Griffiths effects in the choice of the initial state have been proposed [71] to account for the slow dynamics. Note also the recent work [84], which reports the existence of a "slow" phase with respect to the spreading of information in the neighborhood of the Aubry-André ETH/MBL transition.

**Logarithmic growth and MBL regime.**  In the MBL phase, one expects the interactions to slowly entangle the localized particles, resulting in a logarithmic growth of the entanglement entropy [76]. The lower left panel of Fig. 12 displays the entropy as a function of time in log-linear scale, for $h \geq 2.5$. The entropy is seen to grow logarithmically as a function of time at intermediate time scales, after an initial regime, and before reaching its saturation value. The

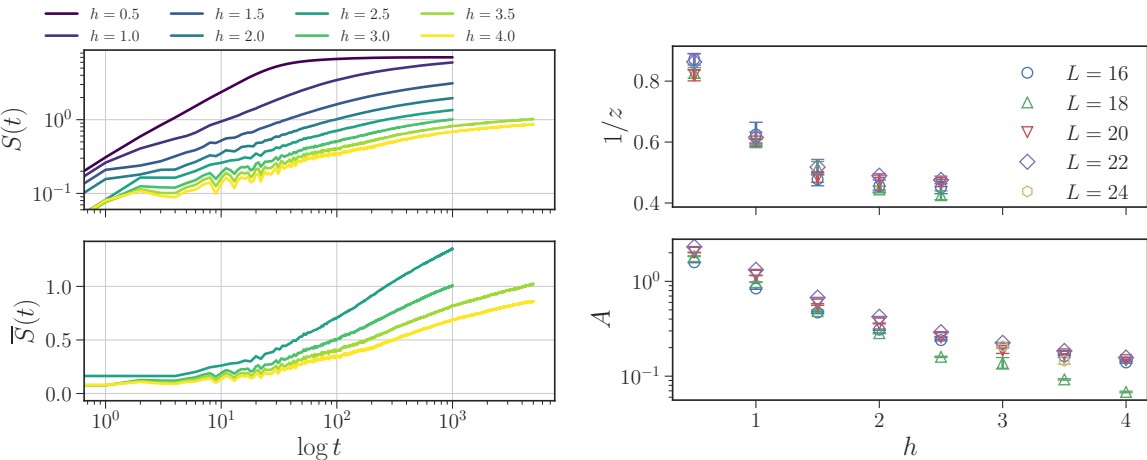

Figure 12: Left: Entanglement as a function of time, in log-log scale (upper panel) and in log-linear scale (lower panel), for the chain of $L = 22$ sites. Right: Best fits of the form $S(t) \propto t^{1/z}$ (upper panel) and $S(t) = A \log t$ (lower panel).

logarithmic growth of the entanglement entropy is not easily seen, and in order to probe it more quantitatively, we display on the lower right panel of Fig. 12 the best fit of the entropy to a logarithmic form $(S(t) = A \log(t) + \text{cst})$ as a function of potential strength, for various system sizes. For $h \lesssim 2.5$, the $A$ prefactor increases linearly with system size, indicating that the entropy does not grow logarithmically with time. For $h \gtrsim 2.5$, we observe a good fit to a logarithmic form, with a size-independent $A$ prefactor[7], in full agreement with the onset of MBL.

In conclusion, the dynamics of high energy product states is consistent with the standard picture of MBL: in the ETH phase, we observe a power-law decay of the density imbalance, together with a power-law growth of the entanglement entropy, while in the MBL phase the imbalance is seen to saturate to a finite value, and the entanglement entropy to grow logarithmically with time. Moreover, the analysis of the dynamics provides us with an estimate of the transition point $(2.5 \lesssim h^* \lesssim 3)$ which is consistent with the one obtained from the analysis of eigenstates properties.

# 7   Conclusion

We have studied a model of interacting spinless fermions subject to the binary Fibonacci potential. In the non-interacting limit, the Fibonacci chain has been extensively studied as a paradigmatic model of quasiperiodicity: exhibiting multifractality, a somehow intermediate behavior between an Anderson localized and a delocalized system.

When interactions are added, we find that the system undergoes a many-body localization transition at a disordered strength $2 \leq h^* \leq 3.5$. Our claim is supported by the exact diagonalization study of static probes (spectral statistics, scaling of the entanglement entropy, statistics of local observables) as well as dynamic ones (entanglement growth and evolution

---

[7]Excluding the outlying $L = 18$ data, whose deviation from the trend we have no explanation for.

of local observables following a quench).

In the ETH phase $(h \leq h^*)$, we observe a sublinear scaling of the natural orbitals, compatible with a survival of the free fermions multifractality. Moreover, we find sub-diffusive dynamics for both imbalance and growth of entanglement entropy, a result which is a bit surprising – but in line with some results on the quasiperiodic Aubry-André model [37, 71, 83, 84] – as the Fibonacci potential is free of rare Griffiths regions, which are usually thought [15, 16] to cause such power-law behaviors.

At large potential strength $(h \geq h^*)$, the Fibonacci MBL exhibits some specific features in the entanglement entropy and local observables, which we relate to the binary and long-range ordered nature of the Fibonacci modulation.

**Future directions** In this article we have studied the Fibonacci chain in the non-interacting limit, and at fixed interaction strength $\Delta = 1$. A possible follow up would be to draw the phase diagram in the $(\Delta, h)$ plane. It would in particular be interesting – albeit technically difficult due to the proximity to integrability – to understand the fate of the transition in the vicinity of the critical line $\Delta = 0$, especially given the instability of the localized phase observed in the Aubry-André model in this limit [86]. We have argued using qualitative mean-field arguments that the non-interacting chain in controlled by a fixed-point unstable to the addition of interactions. To explore this picture, it would be interesting to extend the existing renormalization group approaches to ground-states of aperiodic sequences [22, 23] to high energy, in the lines of [35, 80, 81].

Quasiperiodicity is usually encoded in the hopping terms [22, 48] or in the on-site potentials (this work). Another choice, which may result in new physical properties, is to encode quasiperiodicity in the interaction term.

The Fibonacci sequence studied here belongs to a family of aperiodic sequences [87, 88], which can be classified according to the amplitude of their geometrical fluctuations [89]. Accordingly, the low energy physics of such systems falls into the universality classes of clean or disordered systems, or can become non-universal [23, 24]. The high-energy physics studied here in the context of the Fibonacci chain remains to be explored for the other members of the aperiodic sequences family. Moreover, sequences with stronger geometrical fluctuations such as the one studied in [24] yield more finite-size samples, potentially making their scaling analysis easier.

Many experiments conducted with phonons [90], photons [91], microwaves [92], polaritons [93] and cold atoms [94] explore the properties of non-interacting quantum particles in a quasiperiodic landscape. A cold-atomic experimental setup has been proposed [95] to realize the Fibonacci sequence considered here, which could be used to probe the many-body physics discussed in our work.

## Acknowledgements

We thank Fabian Heidrich-Meissner, Anuradha Jagannathan, Marco Tarzia, Jean-Marc Luck, Cécile Monthus and Marko Žnidarič for fruitful discussions. We thank the anonymous referees for their suggestions and remarks. The shift-invert [13] and time-evolution computations were performed using the PETSc [96], SLEPc [97] and STRUMPACK [98, 99] libraries. Data analysis and plotting was performed using the NumPy, Pandas [100] and Matplotlib [101]

libraries, which are part of the SciPy [102] ecosystem.

**Funding information** This work benefited from the support of the project THERMOLOC ANR-16-CE30-0023-02 of the French National Research Agency (ANR) and by the French Programme Investissements d'Avenir under the program ANR-11-IDEX-0002-02, reference ANR-10-LABX-0037-NEXT. We acknowledge PRACE for awarding access to HLRS's Hazel Hen computer based in Stuttgart, Germany under grant number 2016153659, as well as the use of HPC resources from CALMIP (grants 2017-P0677 and 2018-P0677) and GENCI (Grant 2018-A0030500225).

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
