# Peer review of "Many-body localization in a quasiperiodic Fibonacci chain"

_SciPost Physics_

## Round 2 · Referee Report · Anonymous (Referee 1) · 2018-12-21

Strengths

1 - An interesting study of a model with some unique properties that distinguish it from others typically studied in the context of MBL.
2 - Comprehensive numerical study that covers many different quantities, with convincing arguments backing up the data.
3 - Likely to stimulate further interest in Fibonacci potentials in the MBL community.

Weaknesses

1 - As a result of the large number of different quantities computed by the authors, many are introduced briefly and dispensed with very quickly, making it occasionally hard to follow.
2 - Though the results are solid and interesting, the manuscript would benefit from a stronger punchline as to the significance of the results.

Report

Warnings issued while processing user-supplied markup:

  • Inconsistency: Markdown and reStructuredText syntaxes are mixed. Markdown will be used.
    Add "#coerce:reST" or "#coerce:plain" as the first line of your text to force reStructuredText or no markup.
    You may also contact the helpdesk if the formatting is incorrect and you are unable to edit your text.

This work takes a novel approach to the study of many-body localization by examining a model which is delocalized (and critical) in the non-interacting limit, and becomes localized only when interactions are switched on.

The authors consider a model of interacting fermions in one dimension subject to a quasiperiodic on-site potential of the Fibonacci form. This potential is balanced on a delicate knife-edge, being neither disordered enough to localize the non-interacting system nor smooth enough to entirely delocalize it: this results in the various interesting features of the non-interacting model that are well summarised by the authors.

Using a simple yet persuasive argument, the authors demonstrate that the addition of interactions acts to ‘smooth’ the Fibonacci potential, disrupting this delicate balance. With numerical simulations, the authors go on to demonstrate that the interacting system appears to follow typical MBL phenomenology, i.e. the system is delocalized at small disorder but there is a many-body localization transition at some critical disorder strength. The authors conduct a comprehensive numerical study and present various different quantities which corroborate this claim. As a consequence of the impressive variety of different quantities that the authors consider, some are presented in a rather brief way that relies on readers being already familiar with MBL phenomenology: some small improvements to the discussion would make the manuscript more accessible to a broader audience.

In summary, the authors have given convincing evidence for MBL phenomenology in this model and reviewed the novel features of the Fibonacci potential in a way that serves as an adept introduction of this model to an audience already familiar with MBL. This work likely to stimulate further interest in models with Fibonacci potentials in the MBL community.

I have a variety of minor comments, questions and suggestions which are included in this report. Based on a satisfactory response to these points, I’d be happy to recommend publication in SciPost Physics.

Remarks:

  • The perturbative argument given for the ‘smoothing’ of the Fibonacci potential by interactions is plausible and intuitive. This suggests that, in the language of renormalization group, the non-interacting system is controlled by a fixed point unstable to the addition of interactions. The authors cite a variety of RG works which have examined the low-energy properties of the model, but as a future work, it would be interesting to see if RG methods could examine this model in the context of many-body localization.

  • In Fig. 10, the bar over the I(t) suggests some form of averaging was used, but this isn’t mentioned anywhere in the text - was this data averaged in some way?

  • In Fig. 12, why is the maximum time different for the two largest disorder strengths?

  • In future directions, the authors remark that the fate of the transition in the non-interacting limit presents a technical challenge – what precisely is the challenge that this poses?

Requested changes

1 - The current statement in the abstract that ‘the MBL phase presents specific features’ is a little vague. Could the authors say something more definite here, perhaps giving an example of one?

2 - Can the authors clarify what they mean by the sentence “Finally, discrete disorder does not seem to affect the transition as compared to continuous random potential distributions” (p2)?

3 - Can the authors provide some further argument to clarify why there are L+1 Fibonacci words of length L? Currently this is stated without proof (p4).

4 - The authors comment on ‘large statistical errors’ (p4) as a result of the small number of samples available, due to the intrinsic nature of the model. Can the authors provide any further comment on which features of their study may be most strongly affected by these statistical errors?

5 - The existence of secondary peaks in the density (Fig. 5) is interesting and the explanation is ultimately convincing, though this section is a little convoluted to read. The left panel of Fig. 6, however, clearly demonstrates that the secondary peaks are associated with configurations where neighbouring sites have the same sign: I’d suggest leading with this observation before introducing the magic angle states.

6 - Section 5.3 covers a lot of concepts not previously introduced in the paper, and does so very quickly. A slightly expanded discussion of how multifractality can be quantified using the participation entropy would make this section easier to follow, particularly for those unfamiliar with these quantities.

7 - The use of a logarithmic fit to the growth of entanglement entropy in the ETH phase is explained in a confusing way. Although it makes sense after a few reads, can the authors find a clearer way to explain this?

A few typos I spotted:

1) p2: ‘random potentials distributions’ -> ‘random potential distributions’ 2) p2: ‘recasted’ -> ‘recast’ 3) p3: ‘eigenstates properties’ -> ‘eigenstate properties’ 4) p5: ‘free fermions Fibonacci chain’ -> ‘free fermion Fibonacci chain’ 5) p10: ‘were’ -> ‘where’ 6) p17: ‘quench disordered systems’ -> ‘systems with quenched disorder’ would read better. 7) p7, caption of Fig. 3: the authors say the colour coding is ‘similar’ in both panels, but unless there is some difference, do they mean ‘the same’? 8) p18: ‘disordered strength’ -> ‘disorder strength’

  • validity: high
  • significance: high
  • originality: good
  • clarity: high
  • formatting: excellent
  • grammar: excellent

Author:  Nicolas Macé  on 2019-03-28  [id 477]

(in reply to Report 1 on 2018-12-21)
Category:
answer to question

We thank the referee for his/her positive appreciation of our work, and his/her careful and critical reading.

Remarks

  1. We thank the referee for pointing out that RG studies would be particularly suited to studying this problem. We have added his/her suggestion to the perspectives to our work (third paragraph of Future Directions, p. 18).

  2. The data is averaged both over the available samples and over product states whose average energy is close to the infinite temperature energy. We have clarified this in the caption of Fig. 10.

  3. While full exact diagonalization lets us evolve a state up to arbitrary times, the Krylov-subspace method -- best suited for tackling large system sizes -- used for producing the data presented in Fig. 12 is iterative and requires more effort to compute states up to larger times. For that reason, time evolution is performed for very large times only in the MBL regime, where the slower dynamics renders the extra effort interesting.

  4. In the $\Delta \ll 1$ limit, the system is close to being integrable, making it difficult to study numerically: too small systems will behave as if they were integrable. We added these precisions to the text (first paragraph of Future Directions, p. 18).

Requested changes

  1. The abstract now mentions the fact that the density distribution has extra peaks in the MBL Fibonacci phase.

  2. The sentence on p. 2 has been replaced by "Finally, a discrete disorder distribution can also induce MBL, despite stronger finite size effects observed in the case of binary distributions".

  3. On p. 4, the second paragraph of "Quasiperiodicity, samples of finite size and averaging over realizations" now includes a reference to the proof that Fibonacci sequences have L+1 words of length L.

  4. We have expanded the discussion on the "large statistical errors" in a footnote on p. 4.

  5. On p. 9-10, we have modified the text to make the explanation more understandable, leading with the analysis of Fig. 6 as suggested by the referee.

  6. At the beginning of Sec. 5.3, we have expanded the introduction of the concept of participation entropy, detailing in particular the interpretation of the multifractal dimensions we ultimately compute.

  7. We have rephrased the discussion of the fit in the MBL regime on p. 17 to make it clearer.

Typos

We thank the referee for spotting typos, and have corrected them.

---

## Round 2 · Referee Report · Anonymous (Referee 2) · 2019-1-18

Strengths

1- MBL investigated in a new form quasi-periodic potential where the single particle eigenstates exhibit multifractality. 2- An exhaustive study of observables for large systems amenable to exact diagonalization

Weaknesses

1- The formulation of the problem is relatively less original. 2- Lack of a physical picture for the effects of multifractality on MBL and ETH.

Report

The authors present a detailed study of MBL in a quasiperiodic Fibonacci chain. A quasiperiodic potential is free from Griffiths regions realised in systems with quenched disorder, where local fluctuations produce parts of the chain where the disorder is too weak to be localized. In the absence of such regions, the signatures of MBL are more robust and there are indications that the nature of the MBL-ergodic transition itself changes. The eigenstates of the non-interacting Fibonacci chain exhibit multifractality for all strengths of the potential. These are states which are intermediate between being exponentially localized and extended. Therefore, this system can possibly not be many-body localized or host a form of intermediate behaviour. This question is an interesting one.

In this article, the authors have investigated the spectrum and the eigenstates of the model using exact diagonalization and conclude the existence of a phase transition into an MBL phase. They have studied the transition using a number of static probes such as level statistics, local fermionic density, entanglement entropy and single particle density matrix. They have also studied the dynamics of imbalance and entanglement entropy. The evidence for an MBL phase in this model is very convincing. Although for some quantities like the level statistics (Fig. 4), the finite size scaling near the critical is not very 'clean', largely because the number of different realizations of the quasiperiodic potential is small for the largest system sizes that can be probed using ED. The evidence for the MBL phase is the main result of this work. They also show that the correlated nature of the potential produces certain magic angle states in the MBL phase which leave their imprints in the distribution of the local fermion density and half entanglement entropy. Although this work does not give a physical picture of the nature of MBL in multifractal systems and how it is distinct from conventional randomness, it is nonetheless a useful result and fit for publication.

In section 5.3, the authors present the scaling of the orbital participation entropy with system size for non-interacting and interacting models at low disorder. In the non-interacting case the coefficient of the ln L scaling gives the fractal dimension, which is less than one. On the other hand the authors show the scaling with L in the interacting case and claim this to be a 'multifractal ETH' phase, without any justification. It would be useful if there is a theoretical picture for using this term. Also, it would be desirable for the authors to comment on the relationship of this phase to the non-ergodic, delocalized phase seen in Bethe lattices and random regular graphs (for e.g. Phys. Rev. Lett. 113, 046806 (2014)). In this section, the authors should also include plots of this quantity in the localized phase and comment on any imprints of multifractality that does or does not exist.

In section 6.2 the authors study the growth of entanglement with time for product states. The free Fibonacci chain shows a power law growth, where the power depends on the strength of the quasiperiodicity. At strong quasiperiodicity the 'log-periodic' oscillations are associated with the discrete scale invariance. Based on the discussion in section 3 the discrete scale invariance presumably exists at all strengths of quasiperiodicity, yet the oscillations are absent low 'disorder'. Is this a finite size effect? In non-interacting Anderson localization the saturation value of the entanglement entropy is O(1) and independent of system size. What is the scaling with L of the saturation value of entanglement entropy in the free Fibonacci chain? How does it depend on the fractal dimension?

In the ETH phase, the entanglement entropy grows as a power law in time but the exponent does go to zero at the MBL transition, possibly due to finite-size effects. In the ETH phase does the entanglement entropy saturate to the infinite temperature EE? A plot showing the scaling with L of the saturated value of ETH and MBL phases would provide clarity on this question. In the MBL phase the growth is consistent with being logarithmic in time. The discussion of the fitting procedure is a bit unclear. The distinction between the coefficient of the logarithm between the ETH and MBL phases is not substantial. Could this be because the entanglement growth is much slower in the 'multifractal ETH' phase compared to quenched disorder? A comparison with the binary disorder model studied earlier would be very helpful in clarifying this point. In the plot of A vs h in Fig. 12 the L=18 data deviates quite significantly from the rest of the points. What is causing this?

Overall the manuscript is clearly written, well-structured and motivates the importance of the problem quite well. I recommend it for publishing once the authors address the points raised earlier.

Requested changes

Please refer to the report

  • validity: good
  • significance: good
  • originality: ok
  • clarity: good
  • formatting: good
  • grammar: good

Author:  Nicolas Macé  on 2019-03-28  [id 476]

(in reply to Report 2 on 2019-01-18)
Category:
answer to question

We thank the referee for his/her positive appreciation of our work, his/her careful and critical reading, and his/her ensuing remarks and suggestions.

  1. We agree with the referee that a theoretical understanding of the observed multifractal scaling of the one-particle excitations is desirable. We also think such a picture -- relying for example, as suggested by the referee, on a comparison with the Anderson problem on the Bethe lattice or on random regular graphs -- is difficult to obtain. Indeed, while the behavior of single-particle excitations is reasonnably well-understood from the l-bits picture in the MBL phase (Refs 66-68), analytical approaches are notoriously difficult to implement in the ETH phase -- especially when discrete quasiperiodicity is involved. As for numerical approaches, computing the one-particle density matrix in an unbiased fashion in the ETH phase is a technical challenge, and the exact diagonalization techniques we employed are limited to about 24 sites (this is the state-of-the-art), rendering a careful scaling analysis difficult. Nevertheless, on the length scales that we can probe, our numerics is compatible with a multifractal scaling whose origin remains to be understood. Following the referee's remarks, we have rewritten this part of the text to make it more careful, taking care not to make the claim of a new multifractal phase.

The referee also suggests performing a scaling analysis of the orbital participation entropy in the MBL phase. In the MBL regime, the strength of the quasiperiodic modulation leads to larger oscillations of the average participation as a function of system size, obscuring the scaling behavior. Nevertheless, our data (shown in attached plot 1) appears to be compatible with localization of the orbitals. We mention this in the new version of the manuscript (even though we do not show the data as it would increase the size of our already long manuscript).

  1. The log-periodic oscillations observed in the growth of entanglement entropy are present as soon as $h$ is non-trivial, but are simply difficult to see due to the smallness of their amplitude, that increases continuously with $h$. Our numerics is compatible with the absence of finite-size effects, as we show on plot 2 attached to this reply. We observe that the saturation value of the entanglement entropy is extensive, with a prefactor that decreases continuously as $h$ is increased. We have clarified and enriched the corresponding discussion in the article, following the referee's questions and comments. We believe the question asked by the referee of the relation between the prefactor's decay and the multifractal dimensions of the single-particle orbitals is an interesting yet difficult one. Indeed, simpler problems such as establishing relations between the multifractal dimensions of the orbitals and the diffusion exponents are still open, see eg J. Bellissard, Anomalous transport: results, conjectures and applications to quasicrystals, 2000.

  2. The referee wishes to assess whether the $z > 1$ dynamical exponent we observe in the ETH phase is a finite-size effect or not. S/he suggests analyzing the scaling of the saturation value of the entanglement entropy for that purpose. Following the referee's suggestion, we compare the infinite time entanglement entropy to the Page form $S(t \to \infty, L) = \frac{L}{2} \ln 2 - \frac{1}{2}$ expected for a system thermalized at infinite temperature. We observe that the infinite time entanglement entropy converges to the Page bound as we increase system size, as shown in the $h=1$ case in attached plot 3. Furthermore, after a short-time transient regime, we observe the entanglement entropy to grow as a power-law of time, with a size-independent exponent, as shown in attached plots 4 and 5. In conclusion, the numerics is compatible with a non-triviality of the dynamical exponent. Following the referees suggestion, we have clarified the corresponding discussion in the text. In order again not to increase considerably the lenght of the manuscript, we chose not to include all these plots in the main manuscript (they will nevertheless be available to readers through this reply).

  3. We have clarified the discussion of the fitting procedure in the MBL phase, which is now in our opinion much more readable. We have noted the fact that $L=18$ is an outlier not only with respect to the entanglement growth measure, but for all other estimators. We honestly do not understand the origin of this effect.

Attachment:

attachment_ref2.pdf

---

## Round 2 · Referee Report · Anonymous (Referee 3) · 2019-1-29

Strengths

  1. First study of the infinite temperature dynamical phase diagram in an interacting chain with an aperiodic modulation of the potential
  2. Very thorough numerical study of spectral and dynamical response

Weaknesses

  1. Hard to extract fine structure of the ETH and MBL phases in the model because of larger finite-size effects in the Fibonacci case as opposed to the random or Aubry-Andre case
  2. No analytic estimate of exponent b (defined below Eq. 4) or critical value h_c (see report)

Report

The authors present a very through numerical study of the infinite temperature dynamical phase diagram of the interacting Fibonacci chain. Even though the non-interacting model exhibits critically delocalized single-particle wave functions at all strengths of the potential, the interacting model localizes at larger strength of the potential. The authors present a qualitative argument for this localization and discuss various special features of the MBL phase that are a consequence of the correlated binary disorder.

The study is sound, interesting in the context of localization, and will likely motivate future work. For these reasons, the referee recommends publication. The referee has a few suggestions/questions though; addressing these would significantly strengthen the manuscript.

Big picture questions/suggestions:

1) The authors qualitatively argue that adding interactions smoothens the Fibonacci potential in a mean-field picture. Could this picture be made quantitative? For example, at large h and small Delta, one could start with a typical Slate determinant eigenstate at \Delta=0, compute the average occupation on each site and thus the mean-field, solve for the single-particle wavefunctions with this mean field, feed the new wave functions into the Slater determinant and iterate. The referee expects that this procedure should converge deep in the MBL phase because the eigenstates are approximately product states. If it works, this procedure would allow the authors to extract a modified b (that might be h-dependent) and an estimate for the critical h at small \Delta. It would also be interesting to see how the multi-fractal structure of the starting wave functions is forgotten and how a localization length emerges in this iteration process.

2) The claims about a “multi fractal ETH” phase seem dubious and inconsistent with the qualitative argument for MBL. If the effective exponent b is greater than 1, then the single-particle wave functions are either extended or localized (except at a critical value). So there is really no reason to expect the multi-fractal structure of the b=1 case to show up in the single-particle density matrix of the interacting ETH phase at large L. The structure that the authors are seeing must be a small-size effect.

3) The discussion about the t^z growth of entanglement in the ETH phase is very confusing. As best the referee understands, this type of growth is observed in the non-interacting Fibonacci chain, in random models in the Griffiths region, in the interacting Aubry-Andre model at finite size, as well as in the interacting Fibonacci chain. Do the authors understand the origin of the z>1 growth in the interacting Fibonacci chain? If it is related to the short-distance multi-fractal structure that they observe in the single-particle density matrix, can the exponent in Fig. 9 be related to z in Fig. 12?

Minor questions/suggestions:

1) As another referee suggested, it would be helpful to see the behavior of the saturation value of the entanglement entropy vs L in the ETH phase.

2) As the authors note in the conclusions, other aperiodic sequences have larger geometrical fluctuations (or positive wandering exponents). These sequences might be better for finite-size study of the ETH-MBL transition as the number of independent samples at size L would be larger than in the Fibonacci case.

3) In the top paragraph of Page 6, the authors say that the conductivity at the Fermi level decays as L^{-\alpha}. The authors are presumably talking about T=0, but the relevant properties for this study are at T=infinity. How does the conductivity of the free model behave as a function of L in the infinite T ensemble?

Requested changes

See above

  • validity: top
  • significance: good
  • originality: good
  • clarity: high
  • formatting: excellent
  • grammar: excellent

Author:  Nicolas Macé  on 2019-03-28  [id 475]

(in reply to Report 3 on 2019-01-29)
Category:
answer to question

We thank the referee for his/her thorough report, and his/her ensuing remarks and suggestions.

Big picture questions/suggestions:

  1. Following the referee's suggestion, we have implemented a very naive mean-field procedure whose main conclusion is that the system always localizes, independently of $h$ and $\Delta$. Details of the mean-field procedure we use are as follows: we start by fixing a set of single-particle orbitals, whose Slater determinant is a high-energy many-body eigenstate of the free Fibonacci chain. Then, we perturb the potential as described in Sec. 4.3. We solve the perturbed problem and iterate the procedure. We observe that this naive mean-field converges in the sense that after a few step the density distribution becomes stationary. The converged system is Anderson localized, no matter how small the quasiperiodic field $h$ is. To assess localization, we have computed the half-chain entanglement entropy, and observe that it crosses over from extensive to sub-extensive as a function of system size (see attached plot 1). The crossing occurs at smaller sizes as $h$ is increased. Furthermore, we observe that the fluctuations of the potential become random and uncorrelated (with an average geometrical correlation length of e.g. 10 at $h = 2$). This leads to an effectively infinite exponent $b \to \infty$. Correspondingly, the single-particle wavefunctions are always localized.

There is an important technical caveat to this naive approach: although the procedure converges in the sense that the density distribution becomes stationary, we observe that the local density at a given site keeps fluctuating and never converges. Indeed, at each mean-field step, the potential is perturbed by the density fluctuations for the average value $n = 1/2$, and we find that these fluctuations decay for the free fermions chain at infinite temperature as $1/L^a$ with a potential-dependent $a(h)<1$ ($a(2) \simeq 0.2$). Since the single-particle level spacing decays at least as $1/L$, for $L$ large enough the perturbation induces crossings in the single-particle spectrum. Since at each step we recompute the densities using the same set of single-particle orbitals, some of which have been swapped by the mean-field procedure, we introduce an element of randomness which prevents the mean-field to converge locally. Because of this difficulty, we believe a more involved mean-field procedure is needed to capture not only the localized, but possibly also a putative metallic regime.

We think this avenue of research suggested by the referee is an interesting one, which however goes much beyond the scope of the present study. Indeed the very nature of the mean-field approximation (which was just used here as a suggestive approximate argument) is problematic and it cannot replace the exact calculations that we perform later in the manuscript.

  1. We fully agree with the referee that one needs to be very careful regarding the existence of a "multifractal ETH" phase. As s/he pointed out, our qualitative argument for MBL indicates that the single-particle orbitals at the mean-field level are either localized or extended. However, when interactions are not perturbative, the physics of single-particle orbitals remains to be explored. Our numerics (see detailed reply to Referee 2) is compatible with a multifractal scaling on the lenght scales that we can probe (even though of course we cannot rule out finite-size effects and a different behavior on larger lengths). Following the referee's remark, we have modified our formulation so as to make it more cautious, and in particular not to make the claim of a new multifractal phase.

  2. In the ETH regime, we observe a non-trivial dynamical exponent $t^{(1/z)}$ with $z>1$. This effect is usually attributed to rare regions in random potential models. Here, there are no rare regions since the potential is quasiperiodic. A possible explanation, invoked in the Aubry-André case, is a rare region effect in the configuration of the initial product state. Attached plot 3 illustrates this notion by showing that several product states of equal average energy can have very different dynamics. Alternatively, this could be a finite-size effect, as pointed out by the referee, and also by Referee 2. In order to address this concern, following the referee's suggestion, we have compared the infinite time entanglement entropy to the Page form $S(t \to \infty, L) = \frac{L}{2} \ln 2 - \frac{1}{2}$ expected for a system thermalized at infinite temperature. We observe that the infinite time entanglement entropy converges to the Page bound as we increase system size, as shown in the $h=1$ case in attached plot 2. Furthermore, after a short-time transient regime, we observe the entanglement entropy to grow as a power-law of time, with a size-independent exponent, as shown in attached plots 4 and 5. In conclusion, the numerics is compatible with a non-triviality of the dynamical exponent. Following the referee's remark, we have clarified the corresponding dicussion in the text of the corresponding section.

Finally, the referee asks whether the dynamical exponent could be related to the observed multifractal scaling of the single-particle orbitals or not. We think this question is a very interesting, but also very difficult one. Indeed, simpler problems such as establishing relations between the multifractal dimensions of the orbitals and the diffusion exponents in the free fermions case are still open, see eg J. Bellissard, Anomalous transport: results, conjectures and applications to quasicrystals, 2000.

Minor questions/suggestions:

  1. We thank the referee for this suggestion. We have looked at the saturation entanglement entropy in the ETH phase (see attached plot 2, and see point 3. of the above discussion). We have added the corresponding informations in the paragraph "Interacting chain: Power-law growth in the ETH phase" of the text.

  2. We thank the referee for pointing out that other non-periodic sequences may have more finite-size words to average about. We have included this remark in "Future directions" paragraph.

  3. We thank the referee for his/her remark. Indeed, we are referring to the $T=0$ conductivity. Given the fate of the other studied observables (one-particle density matrix, half-chain entanglement entropy), we believe the conductivity in the $T \to \infty$ ensemble will also decay as a power-law of system size, with an $h$ dependent exponent. We have added this remark to the text (p. 5, Sec. 3).

Attachment:

attachment.pdf

---

## Round 3 · Author Response

We thank the referees for careful reading our manuscript and the helpful comments/questions. We have replied to all referees' requests (detailed responses can be found in the comments to the reports). We additionally provide a full list of changes for completeness. We hope that after these improvements you will judge our manuscript suitable for publication.

---

## Round 3 · List of Changes

1. The abstract now mentions the fact that the density distribution has extra peaks in the MBL Fibonacci phase.

2. The sentence on p. 2 has been replaced by "Finally, a discrete disorder distribution can also induce MBL, despite stronger finite size effects observed in the case of binary distributions".

3. On p. 4, the second paragraph of "Quasiperiodicity, samples of finite size and averaging over realizations" now includes a reference to the proof that Fibonacci sequences have L+1 words of length L.

4. We have expanded the discussion on the "large statistical errors" in a footnote on p. 4.

5. On p. 9-10, we have modified the text to make the explanation more understandable, leading with the analysis of Fig. 6 as suggested by the referee.

6. At the beginning of Sec. 5.3, we have expanded the introduction of the concept of participation entropy, detailing in particular the interpretation of the multifractal dimensions we ultimately compute.

7. We have rephrased the discussion of the fit in the MBL regime on p. 17 to make it clearer.

8. We have rewritten Sec. 5.3 to make it more careful, taking care not to make the claim of a new multifractal phase. We also added extra informations on the localisation of the orbitals at larger disorder.

9. We have clarified and enriched the discussion on entanglement growth in the free fermions model (Sec. 6.2).

10. We have clarified and added new details to the interacting chain dynamical exponent discussion (Sec. 6.2). We have also clarified the fitting procedure discussion (same Sec.).

11. We have expanded the discussion of the possible origin of the anomalous dynamical exponent (Sec. 6.2). We also added informations about the entanglement saturation value in the same section.

12. We have included the referee's remark about other deterministic geometries in the "Future directions" part.

13. We have made our statement about the free fermion's conductivity more precise (p. 5, Sec. 3).

---

## Editorial Decision

ontology_/_topics